# How Much Pressure Do Students Need to Achieve Good Grades?—The Relevance of Autonomy Support and School-Related Pressure for Vitality, Contentment with, and Performance in School

**Daniela Martinek** [1],*[ID]**, Joerg Zumbach** [2] **and Matteo Carmignola** [2],*[ID]

[1] Rectorate, Salzburg University of Education, 5020 Salzburg, Austria
[2] Department of Educational Science, School of Education, University of Salzburg, 5020 Salzburg, Austria; joerg.zumbach@plus.ac.at
\* Correspondence: daniela.martinek@phsalzburg.at (D.M.); matteo.carmignola@plus.ac.at (M.C.)

**Abstract:** This study investigates, based on Deci and Ryan's self-determination theory, how autonomy support and school-related pressure are associated with students' vitality, their contentment with and academic performance in school, and whether feeling related to teachers and feeling competent mediate these relations. In total, 812 secondary school students participated in this questionnaire-based survey. Perceived autonomy support was positively related while school-related pressure was negatively related with vitality and contentment. Relations were partially mediated by relatedness to teachers and perceived competence. In sum, this study provides insight into how autonomy support contributes not only to better academic achievement but also to students feeling vital in school and experiencing contentment with school environments. Moreover, the results emphasize that pressure is not only irrelevant for academic performance, but rather, detrimental for students' perceptions in school. The practical implications imply that teachers should be trained to avoid unnecessary coercion and to strengthen their abilities in supporting their students' autonomy. This contributes to make school a productive and enjoyable environment for learners and teachers alike.

**Keywords:** autonomy support; school-related pressure; contentment with school; vitality; academic performance

## 1. The Relevance of Autonomy Support and School-Related Pressure for Vitality, Contentment with, and Performance in School

Formal education in schools is an important aspect in the development of adolescents and influences their future opportunities [1]. In order to support the quality of this formal school education, educational research has developed several approaches to improve students' engagement [2], learning outcomes [3], and successful learning strategies [4]. Nevertheless, such approaches tend to focus on a process–product paradigm that is a mere top–down process. Here, self-determination theory (SDT [5]) enriches traditional top–down instructional strategies by additionally taking the needs and perspectives of students into account [6,7]. Instructional concepts following SDT stress the role of students in building supportive learning environments that contribute to the satisfaction of psychological needs and encourage active involvement in learning processes [8].

An extensive line of research has documented the benefits of autonomy support in schools (cf. [9–12]). Longitudinal research by [13] demonstrated that teacher autonomy support reduces adolescent anxiety and depression. Reeve et al. [14] found in an intervention study that autonomy-supportive teaching behavior, which proved to be trainable [15,16], enhanced student engagement in class. Ferguson et al. [9] reported that cross-national differences in school and life satisfaction were partially mediated by adolescents' perceptions of autonomy support.

Our research aimed to stress the importance of autonomy support from the perspective of students by linking students' autonomy perceptions with desirable outcome variables in school. Respecting the need for autonomy implies that students can learn within autonomy-supportive environments that contribute to their psychological growth and well-being [17]. It also implies avoiding controlling teaching behavior and coercive strategies that can put students under pressure [18]. A large body of empirical research has provided a differentiated picture of perceptions of self-determination and a variety of autonomy-supportive strategies [19–21], and of the impact of controlling teacher behavior [22,23]. In a recent study, Kaplan [7] found that teachers' conditional negative regard was an aspect of controlling teaching behavior that was negatively associated with need satisfaction.

Although there is evidence concerning the advantage of autonomy support, teachers may struggle to abstain from controlling teaching behavior. They have often argued that without pressure, their students do not reach their full potential [19]. In applied research, there has been a lack of studies focusing on school-related pressure from the learners' perspective. This paper examines students' perceptions of pressure in school and perceived autonomy support, and analyzes how these aspects relate to students' vitality, their contentment with, and their performance in school, bearing in mind potential mediators, such as relatedness to teachers and perceived competence.

## 1.1. Autonomy and Pressure in School

The experience of self-determination, which within SDT is also called autonomy, can be divided into two different motivational experiences: intrinsic motivation and autonomous forms of extrinsic motivation [12]. If students have the chance to pursue their interests and their internal perceived locus of causality [5,24] in school, and if they have the opportunity to seek optimal challenges for learning, they experience higher satisfaction of their needs for autonomy and competence. Nonetheless, autonomy can also be experienced if the impulse to act derives from the teacher, as long as students can combine the aims and/or the requested behavior with their inner self—referring to their values, interests, and goals. In this case, autonomous forms of extrinsic motivation are stimulated. Experiencing autonomy in school and feeling related to teachers play an influential role in this respect [25,26]. Teachers who abstain from coercive and pressurizing strategies and focus on autonomy-supportive teaching styles use informational language, such as positive verbal feedback, provide meaningful choices for students, show empathy and acceptance, and try to activate the inner motivational resources of their students [20,26]. Controlling teaching behavior based on pressure and coercion leads to controlled forms of motivation [27], meaning that students engage in school-related work because they try to avoid feelings of guilt, shame, or anxiety, or because they feel forced to do so, e.g., by rewards or (threats of) punishment. School pressure here is not regarded as the trigger for controlling behavior, but rather a result from teachers' intentions to control students. Thus, students are assumed to perceive this behavior as controlling, which is subsequently negatively related to perceived autonomy.

Controlled self-regulation is detrimental for psychological need satisfaction and in-depth learning. On the contrary, autonomy support contributes to maintaining and developing intrinsic motivation and autonomous self-regulation, and leads to a positive classroom atmosphere and higher academic achievement [12,23]. In an autonomy-supportive environment, students' perspectives, thoughts, feelings, and actions are acknowledged and accepted [23]. Specifically, learners' perceptions of autonomy are supported by nurturing their inner motivational sources [11], providing a supportive structure [28], offering rationales [19,29], acknowledging and accepting expressions of negative affect [30], and relying on non-controlling language and behavior [16,31]. Autonomy-supportive learning environments have been shown to lead to favorable educational outcomes, such as autonomous motivation, well-being, and increased academic performance, mostly measured by self-reports of GPA [32–36].

Teachers who show controlling behavior in the classroom tend to put their students under pressure and decrease the students' perceptions of autonomy [37,38]. Students feel under pressure in school if they are urged to adopt their teacher's perspectives by the teacher influencing learners' thoughts, feelings, or actions, and pressuring learners to think, feel, or behave in a specific way [23]. Specifically, in controlled learning environments, external sources of motivation, such as surveillance or rewards, are used [26]. Learners are confronted with pressure-inducing language and behavior, impatience, the neglect of rationales, and their complaints and expressions of negative affect are ignored [11,17,18,30,39]. Controlled learning environments in schools are not conducive to the satisfaction of psychological needs; they increase perceived pressure and lead to controlled motivation [40].

Given the benefits of autonomy support opposed to pressure, one may question why it is difficult for many teachers to let go of unnecessary control in the classroom. There are two possible reasons: On the one hand, teachers report that they are often not aware of the variety of autonomy-supportive strategies [41,42]. On the other hand, if teachers feel under pressure themselves, they tend to pass this pressure on to their students [43]. In addition, teachers often think that pressure is essential to make (some of) their students work [23]. Although prior studies focused on what causes controlling teaching behavior in school systems [23,44], there has been little research emphasizing the consequences of perceived pressure from the students' point of view, and whether it substantially contributes to student performance. Therefore, this study did not only analyze the relation between school-related pressure and performance, but it also included influential aspects of well-being, such as experiencing vitality and contentment with school.

### 1.2. Mediating Effects of Relatedness and Perceived Competence

Perceived autonomy support has been shown to be directly related to the satisfaction of basic psychological needs, psychological well-being, and academic achievement [11,35,38]. This research focused on the relations between perceived autonomy and the other two basic psychological needs [5]: the need for competence and the need for relatedness. Contrary to other psychological approaches, which focus on individual differences of learners acquired through learning or socialization processes, the satisfaction of the basic psychological needs of autonomy, competence, and relatedness has been shown to be universal across cultures and ages and is associated with optimal development, integrity, and psychological and physical well-being [12,45]. According to SDT, all three basic psychosocial needs are associated with favorable outcomes [38]. On the other hand, each need pertains to a distinct set of experiences, and the strength of association with related variables differs according to the characteristics of the outcomes [34,46]. Correspondingly, research in SDT has focused on the identification of different sets of strategies to enhance a specific need (e.g., [47]), as well as examining mediating for each need [48].

The aim of this study was to examine the mediating effects of perceived competence and relatedness on the relation between perceived autonomy and school-related pressure on the one hand, and well-being and performance variables on the other hand. In particular, SDT states that perceived autonomy is conducive to the satisfaction of the need for competence, which in turn may have an impact on school performance [32,49–51]. Students who experience autonomy and feel competent in school perceive their work on school tasks to be effective and experience opportunities to explore their personal capacities [26]. This subsequently leads to proactive commitment and improved academic performance [51–53]. If a student's need for autonomy is suppressed (e.g., by perceived pressure in school), this is likely to have a detrimental effect on perceived competence and in turn on school performance [54]. Moreover, our study analyzed if and how school-related pressure was directly linked with performance outcomes.

With respect to the factors fostering a positive motivational orientation towards learning, students' well-being in school is as important as their academic achievement [55], which is why vitality and contentment with school, as two factors related to well-being, were integrated into this study. The satisfaction of the need for relatedness as one of the

three basic psychological needs refers to feeling connected to others (here, feeling connected to teachers in school) and experiencing a sense of belonging in relation to other people and the school community [26]. In autonomy-supportive environments, students feel a close connection to their teachers because they perceive them as people who care and are available whenever advice, support, or help is needed [56,57], which in turn contributes to learners' vitality and contentment with school. Whereas feeling connected to educators in an autonomy-supportive setting is likely to contribute to students' well-being, pressure-inducing environments are negatively associated with relatedness, and in turn, vitality and contentment [58]. Vitality is defined as having mental and physical energy, and includes the experience of enthusiasm, aliveness, and energy available to the self [59]. Students in autonomy-supportive environments in school experience more vitality [52,60]. In line with prior research [60,61], our study suggested that experiencing need satisfaction may play a mediating role between perceived autonomy and vitality. These relations are likely to reverse in pressure-inducing environments as psychological needs are frustrated and the experience of full functioning as expressed by vitality is impaired [10,62–64].

*1.3. Present Research*

The main aim of this study was to investigate how perceived autonomy support and school-related pressure were associated with feeling vital in school, experiencing contentment with school, and school performance. Connecting to prior research [11,32–34,65], perceived autonomy support was expected to be positively related with vitality, contentment, and performance, whereas school-related pressure was expected to be negatively related with these variables [17,18,39,40]. Furthermore, our study explored the role of relatedness to teachers and the experience of competence in school. In line with other studies [48,49,52,66], feeling competent was expected to be strongly related with school performance and positively associated with vitality and contentment. Experiencing teachers as mentors who care and feeling related to these teachers were expected to contribute to vitality and contentment [5,23,56], whereas the relation to school performance was expected to be insignificant [34].

The following hypotheses were tested in the study (see also Figure 1):

- Perceived autonomy support was positively related to the feeling of relatedness to teachers, perceived competence, vitality, contentment in school, and school performance.
- School-related pressure reported negative associations with the feeling of relatedness to teachers, perceived competence, vitality, contentment in school, and school performance.
- The impact of autonomy support on all three outcomes was expected to be increased by a complimentary mediation effect with the perceived competence. Additionally, a complementary mediation effect was expected for autonomy support with the variable for relatedness towards vitality and the contentment with school, while no link with school performance was expected. The interaction with the mediating variables towards school-related pressure was expected to result in competitive mediations buffering the negative effect of pressure on the outcome variables.

The analysis was conducted in order to examine the relations between perceived autonomy support and school-related pressure towards vitality, contentment with school, and school performance. Additionally, the model investigated whether this relationship was mediated by relatedness to teachers and perceived competence [48,66]. The combination of perceived autonomy support and school-related pressure, relying on the perception of secondary school pupils, offers a new perspective on the consequences of need-thwarting learning environments. Additionally, the study added to previous research by addressing performance-oriented outcome measures (GPA) and non-cognitive variables (vitality and contentment), which were simultaneously relevant for developing a positive attitude towards (life-long) learning [9,34].

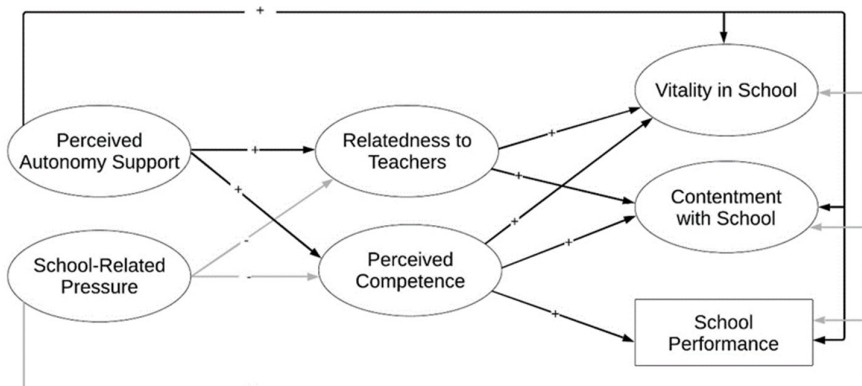

**Figure 1.** Hypothetical model. Note: hypothetical model of relations between perceived autonomy support and school-related pressure with the mediating effects of relatedness to teachers, perceived competence on vitality, contentment with school, and school performance. Black paths indicate positive, while grey lines depict negative expected regression coefficients.

## 2. Methods

### 2.1. Participant and Procedure

A total of 812 students from Austria participated in this study. Their ages ranged from 13 to 20 years-old ($M$ = 16.00, $SD$ = 1.72); 62.6% were women and 37.2% were men, while two participants (0.2%) indicated 'gender diverse'. All participants attended secondary school: 19% were in lower secondary school (grades 5 to 8), 40% were in higher secondary school (grades 9 to 12), and 41% were in higher vocational schools (grades 9 to 13). The sample was representative for the Austrian secondary school system [67]. Participants were asked to voluntarily complete a paper-and-pencil questionnaire in out-of-school settings, which allowed us to analyze the data without controlling for a class- or school-based nested structure. The survey took about 30 min to complete and was carried out following participants' informed consent (further details in the Ethics Statement at the end of the paper).

### 2.2. Measures

For all scales, the survey presented a 5-point Likert scale ranging from 1 = completely disagree to 5 = fully agree.

#### 2.2.1. Perceived Autonomy Support

Perceived autonomy support in schools was measured using an adjusted scale based on the work of Reeve et al. [68]. The scale consisted of nine items measuring the perceived locus of causality (e.g., "I feel I can follow my own aims and goals in lectures"), relevance (e.g., "The content of lectures is relevant to me"), and perceived choice (e.g., "I feel I can explore my own approaches in lectures"; Cronbach's alpha $\alpha$ = 0.82, McDonald's omega $\omega$ = 0.83).

#### 2.2.2. School-Related Pressure

To assess school-related pressure, seven items retrieved from the pressure and tension sub-scales of the Intrinsic Motivation Inventory [69,70] were slightly adapted for the context of schools (e.g., "I feel pressured to do things at school that I wouldn't choose to do"; $\alpha$ = 0.79, $\omega$ = 0.80).

#### 2.2.3. Relatedness to Teachers

A scale consisting of six items based on the items for relatedness from the Basic Psychological Needs Satisfaction Scale [71] and the Learning Climate Questionnaire [32] was used to measure relatedness to teachers (e.g., "My teachers in school are friendly"; $\alpha$ = 0.69, $\omega$ = 0.72).

### 2.2.4. Perceived Competence

Perceived competence was measured using five contextualized items (e.g., "I feel confident in my ability to learn in school") based on the Perceived Competence for Learning Scale [72,73] ($\alpha = 0.87$, $\omega = 0.88$).

### 2.2.5. Vitality in School

In order to assess vitality in school, a German version of the Subjective Vitality Scale [59] with seven items (e.g., "I feel alive and vital in school") was applied ($\alpha = 0.89$, $\omega = 0.90$).

### 2.2.6. Contentment with School

Participants rated their overall contentment with school by four items (e.g., "I enjoy going to school") retrieved from a subscale of a larger questionnaire on well-being in school [74]. The scale achieved good reliability measurements ($\alpha = 0.82$, $\omega = 0.83$).

### 2.2.7. GPA

As an indicator of school performance, GPA was computed based on students' grades from their last report cards for the three main subjects (German, English, and Mathematics). According to the Austrian grading system, the values ranged from 1 = very good to 5 = fail; these were re-coded to increase the interpretability by the readers.

### *2.3. Data Analysis*

The main aim of this study was to analyze the relations between perceived autonomy support and school-related pressure on the one hand, and vitality, contentment with, and performance in school on the other hand, as well as the mediating roles of relatedness to teachers and perceived competence (see Section 1.3 for the hypothesis model). As a preliminary analysis, the correlations among all variables were examined. Subsequently, we performed an analysis of the latent relations between the constructs using structural equation modeling [75]. For the statistical analyses, a package lavaan [76] for the statistical software framework R was adopted. Next to chi-square statistics [75], TLI, CFI, RMSEA, and SRMR were used to estimate model fit [77]. Additionally, the mediating effects were estimated by a bootstrapping approach, as recommended for small mediating effects [78], and interpreted according to the taxonomy of mediations presented by Zhao et al. [79].

### 3. Results

On average, students reported overall moderate scores with regard to all variables (see Table 1). Perceived competence ($M = 3.84$, $SD = 0.72$) and overall contentment with school ($M = 3.67$, $SD = 0.78$) were slightly higher, while vitality in school ($M = 2.83$, $SD = 0.72$) was below average in comparison with the other variables, as shown in Table 1. The lowest correlation was observed between performance and perceived autonomy support ($r = 0.15$, $p < 0.001$). School-related pressure correlated negatively with all the other variables.

**Table 1.** Means, standard deviations, and correlations among all variables in Study 1.

| Variables | *M* | *SD* | 2. | 3. | 4. | 5. | 6. | 7. |
|---|---|---|---|---|---|---|---|---|
| 1. Perceived Autonomy Support | 3.14 | 0.67 | −0.53 | 0.41 | 0.58 | 0.58 | 0.65 | 0.15 |
| 2. School-related Pressure | 2.98 | 0.80 | | −0.40 | −0.51 | −0.56 | −0.55 | −0.23 |
| 3. Relatedness to Teachers | 3.40 | 0.72 | | | 0.37 | 0.42 | 0.47 | 0.22 |
| 4. Perceived Competence | 3.84 | 0.72 | | | | 0.48 | 0.51 | 0.63 |
| 5. Vitality in School | 2.83 | 0.88 | | | | | 0.63 | 0.22 |
| 6. Contentment with School | 3.67 | 0.78 | | | | | | 0.24 |
| 7. GPA | 3.57 | 0.89 | | | | | | |

Note. All correlations are significant at the level *p* < 0.001; measured on a 5-point scale.

To analyze the relation of perceived autonomy support and school-related pressure with vitality, contentment, and performance, with the mediating effects of relatedness to teachers and perceived competence, a structural equation model (SEM, Figure 2) was employed. The SEM (see Figure 2) reported a good model fit ($\chi2$ = 1765.38, *df* = 500, RMSEA = 0.06, SRMR = 0.06, TLI = 0.90, and CFI = 0.91).

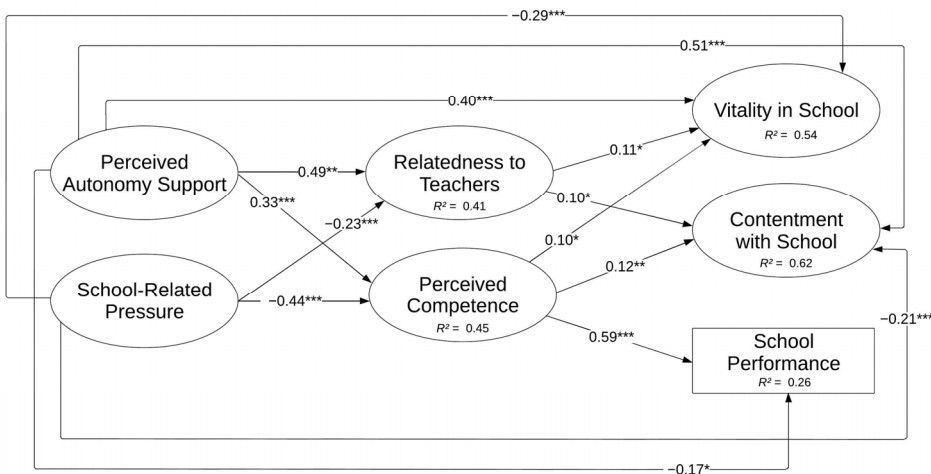

**Figure 2.** Structural equation model. Note: * = *p* < 0.05, ** = *p* < 0.01, *** = *p* < 0.001.

Perceived autonomy support was positively linked with relatedness to teachers ($\beta$ = 0.49, *p* < 0.01) and perceived competence ($\beta$ = 0.33, *p* < 0.001), as well as through distal paths to vitality in school ($\beta$ = 0.40, *p* < 0.001) and contentment with school ($\beta$ = 0.51, *p* < 0.001). The path towards school performance indicated a negative coefficient ($\beta$ = −0.17, *p* < 0.05).

The SEM reported a negative link between school-related pressure and relatedness to teachers ($\beta$ = −0.23, *p* < 0.001) and perceived competence ($\beta$ = −0.44, *p* < 0.001). Furthermore, the paths towards vitality in school ($\beta$ = −0.29, *p* < 0.001) and contentment with school ($\beta$ = −0.21, *p* < 0.001) reported negative coefficients. School-related pressure had no significant regression effect on school performance.

The covariances between the latent constructs were significant for perceived autonomy support and school-related pressure (*r* = −0.50, *p* < 0.001), between vitality and school contentment (*r* = 0.37, *p* < 0.001), as well as contentment and school performance (*r* = 0.15, *p* < 0.01).

Additionally, all interactions were tested for mediating effects; the regression effect of perceived autonomy support on vitality and on contentment with school was supported by a complementary mediation through perceived competence. The indirect effect was significant for both interactions ($\beta_{indirect}$ = 0.03 and 0.04, *p* < 0.05), resulting in an increased total regression effect of $\beta_{total}$ = 0.43 on vitality and $\beta_{total}$ = 0.55 on contentment (both *p* < 0.001). Furthermore, the impact of school-related pressure reported a significant interaction with perceived competence; both the indirect effect in regard to vitality ($\beta_{indirect}$ = −0.05, *p* < 0.05), as well as towards contentment with school ($\beta_{indirect}$ = 0.04, *p* < 0.05), increased the overall negative effect of school-related pressure ($\beta_{total}$ = −0.33 and −0.26, *p* < 0.001) on both outcome variables.

In total, the SEM explained between 39% and 44% of variance for the mediating variables, while the explained variance for the outcome variables was partially higher ($R^2$ was between 0.26 and 0.62).

## 4. Discussion

Whereas the consequences of observed controlling teaching behavior have been studied [43,44], school-related pressure and perceived autonomy support have less often been combined in empirical research. The strength of the present study was that on the one hand,

school-related pressure was assessed from the students' point of view and not by observed or rated teaching behavior. On the other hand, basic psychological needs were separately analyzed, allowing for a deeper understanding of relations to relevant outcome variables. This study investigated the relations of perceived autonomy support and school-related pressure (independent variables) with vitality, contentment, and performance (outcome variables), and the mediating effects of relatedness to teachers and perceived competence for a sample of secondary school students. The model predicted up to 62% of the variance of the outcome variables and reported several regression effects of the independent variables on the mediating and outcome variables. Perceived autonomy showed both proximal and distal effects towards all variables and was associated with higher levels of relatedness and competence, as well as increased vitality and contentment with school [80,81]. On the other hand, school-related pressure was mainly linked with lower perceived competence, lower relatedness, lower vitality, and a reduced contentment with school [82]. For school performance, measured by GPA, the model reported a positive and large regression effect from perceived competence, but also a negative direct path from perceived autonomy support, despite two positive coefficients between autonomy and competence, and competence and school performance. This unexpected result is addressed in the limitations below, as the comparison with the correlation matrix may suggest a suppression effect. Furthermore, the SEM reported no direct link between school-related pressure and school performance, but rather, several negative outcomes on other variables. The reported regression effects were supported by a few complementary mediations that increased the total effect of the perceived autonomy and the school-related pressure on vitality and contentment in school.

Providing autonomy for students in school is a core approach to establishing a motivating and stimulating learning environment and climate [14]. According to SDT [34], autonomy support contributes to students' need satisfaction, students' well-being, and due to more proactive learning behavior, their academic performance. In line with prior research (cp. [26,51]), the results of this study showed that from the students' perspective, autonomy support is positively linked with feeling related to teachers and feeling competent concerning challenges in school, and it also contributes directly to the experience of vitality and contentment. An autonomy-supportive climate is beneficial for creating a positive relationship with teachers that ideally contributes to a positive interdependence and an atmosphere of reciprocal trust and respect among students and teachers [83]. A major reason for this is that this form of support and trust meets basic psychological needs, as stated in SDT [84]. If students experience autonomy, they act proactively and feel more competent when corresponding skills and abilities are available [53], which results in better academic performance. Thus, experienced autonomy support mediated by students' perceptions of their own competence contributes to increased performance in schools as assessed by standard grading.

Perceived autonomy support was exclusively positively associated with positive outcomes in this study with one exception. Although mediated by perceived competence, autonomy support contributed significantly to academic performance; unexpectedly, the direct relation between perceived autonomy support and grades in the structural equation model was negative. However, the coefficient was rather small and the moderate-to-high covariance between autonomy and competence may have caused a suppression effect because the bivariate correlation between competence and GPA (see Table 1) is positive. Nevertheless, there is a small chance that a lack of structure could explain this relation. At the same time, autonomy support required a clear structure [85], which was not assessed in this study. The competences of students were not assessed. One can conclude that students can experience autonomy in school, yet due to their competence levels, the relation to their grades may vary. This is a finding to be addressed in future research.

Nonetheless, inducing pressure was not associated with better performance and school-related pressure was not directly linked with school performance in our study. Learning environments where teachers do not support students' autonomy and act highly controlling are likely to alienate students from teachers and are not conducive for need

satisfaction, well-being, and performance [71]. The current study identified perceived school-related pressure as an obstacle to experiencing a stimulating learning environment. School-related pressure was associated with lower levels of perceived competence and a reduced experience of positive student–teacher relationships. Pressure lowers the degree of autonomy with which students can develop or thrive within the academic environment [86] and is therefore detrimental to realizing students' abilities and developing trust in these abilities. Corresponding with prior research [7], the data showed that experiencing school-related pressure reduced levels of vitality and contentment.

To sum, pressure was not only an obstacle to better academic performance, but it also significantly reduced all favorable outcome variables. School-related pressure and coercion were not only ineffective in improving performance, but also had the detrimental effect of reducing students' need satisfaction and their well-being in school.

*Limitations*

There were some limitations of this study. First, the sample was obtained through a snowball approach, and despite being representative in regard to the school types [67], other sampling effects could not fully be controlled. Second, in addition to GPA, all constructs were assessed trough self-report scales; this may have reduced the external validity for the independent variables where only perceived autonomy support and experienced pressure were assessed. Here, observational studies on autonomy-supportive and pressuring behaviors in school [11] may have increased the validity of the results. Furthermore, the context of the survey did not reflect specific programs or lesson designs by teachers who already provided autonomy support in school, but rather presented a cross-sectional sample among different schools, school types, curricula, and teachers. The cross-sectional design did not allow for causality to the predictors to be inferred, which could only be achieved by experimental or measurement-intense longitudinal designs, which additionally could contribute to identify reciprocal patterns between students' engagement, academic outcomes, and controlling teaching behavior [44,86].

With regard to the analysis procedure, the intercorrelations of the measures could reduce a robust determination of the regression effects, as the unexpected negative regression path between autonomy and GPA could suggest a negative suppression [75] (p. 37). For this, a stepwise approach in the model specification was employed to ensure the interpretability of the results.

**5. Conclusions**

In line with SDT, the experience of autonomy was a key element in education [87] and practitioners are advised to avoid unnecessary pressure in schools [10,11,50]. Following a bottom–up perspective, we recommend that teachers integrate students' perspectives on autonomy support and pressure-related factors and use this information to (further) develop the design of learning environments in schools. In an autonomy-supportive learning environment, teachers ideally provide learners with a supportive structure, challenging tasks, and informative feedback. This enables students to develop their competence and establishes a positive interdependence among teachers and students that reduces pressure and allows students to actively participate and experience vitality in school [29,59]. The positive consequences of such learning environments not only increase students' contentment with school, but also their academic performance and well-being. We recommend analyzing existing learning environments in schools to further enhance the experience of autonomy in school. The focus ought to be on implementing autonomy-supportive strategies on the one hand and reducing excessive pressure and coercion on the other hand, in order to consider both the bright and the dark sides of motivation in teaching [88].

**Author Contributions:** Conceptualization, D.M., J.Z. and M.C.; Methodology, D.M., J.Z. and M.C.; Analysis, D.M., J.Z. and M.C.; Writing—original draft, D.M., J.Z. and M.C. All authors have read and agreed to the published version of the manuscript.

**Funding:** The authors acknowledge the Open Access Funding by the University of Salzburg.

**Institutional Review Board Statement:** The work presented in this paper was carried out in accordance with the APA ethical standards. Ethical review and approval were waived for this study, as the survey is not linked to any risks to participants and did not entail a collection of personal or sensitive data.

**Informed Consent Statement:** Following a preliminary written and oral information, all participants in our study provided their written informed consent in accordance with the Declaration of Helsinki and were granted the right to withdraw from the survey/interview for any reason, without penalty.

**Data Availability Statement:** The datasets generated during and analyzed in the current study are not publicly available due to further, ongoing research projects, but are available from the corresponding author upon reasonable request.

**Conflicts of Interest:** On behalf of all authors, the corresponding author states that there is no conflict of interest.

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
