# Peer review of "How Much Pressure Do Students Need to Achieve Good Grades?—The Relevance of Autonomy Support and School-Related Pressure for Vitality, Contentment with, and Performance in School"

_education, doi:10.3390/educsci12080510_

Round 1

Reviewer 1 Report

The paper is an excellent work to confirm current studies that advocate student autonomy as a key element for successful education over pressure sustained approaches.

It is a fluid reading manuscript, in which each of the stages of the investigation process is clearly presented. The general framework makes it possible to understand the rationale for the variables proposed for the model, resulting in the formulation of the hypotheses tested by the model. The methodology and methods used are well described and substantiated, from the definition of the sample to the scales adopted. References are extensive, as is desirable in structural equation models.

It remains to congratulate the authors for their research work and encourage the continuation of the studies.

Author Response

Thank you for your review of our work and your help in the revision of the manuscript.

Kind regard, The Authors

Reviewer 2 Report

Clarify the aim of the study. You wrote:

This research here aims to stress the importance of autonomy support from the per- 45 spective of students by linking students’ autonomy perceptions to desirable outcome var- 46 iables in school. 

The aim of this study is to examine the mediating effects of perceived competence 127 and relatedness on the relation between perceived autonomy and school-related pressure 128 on the one hand and well-being and performance variables on the other hand.

The main aim of this study is to investigate how perceived autonomy support and 160 school-related pressure are associated with feeling vital in school, experiencing content- 161 ment with school and school performance.

The main aim in this study was to analyze the relations between perceived autonomy 244 support and school-related pressure on the one hand, and vitality, contentment with and 245 performance in school, on the other hand, as well as the mediating roles of relatedness to 246 teachers and perceived competence.

Please provide a statement of third-party approval that you secured to conduct this study (e.g., Institutional Review Board for the Protection of Human Subjects) or if your local context does not require such oversight, then please indicate this and describe how you ensured ethical research practice to protect participants’ safety, privacy, and confidentiality. If the study was deemed to be exempted or excluded from IRB review, please make note of it.

Please add an about the researcher section so you tell us more about you as the researcher(s) and your connection to this study. How does this align with personal interests, professional work, etc., to help the reader place you directly in the center of your work?

Expand your methodology section and please add more recent support to your literature review. All but one are more than 5 years old. 

Author Response

Thank you for your review of our work and your precious feedback, which contributed to the revision of the manuscript.

Following your recommendation, we provided additional information regarding the research ethics and the waiver for ethical board approval. Additionally, a brief note was included in the narrative with a reference to the ethics statement at the end of the manuscript. The Authors’ bio was only included in the covering letter, as it would have jeopardized the blind review: In the revision, we are happy to include three brief statements on the authors.

The methodology follows APA’s Journal Article Reporting Standards (JARS). In order to better integrate the theoretical work on the hypothesis with the analysis strategy, we decided to insert a reference to link the analysis with the hypothesis model.

As you correctly noted, some literature might seem outdated in regard to the year of publication: As we aimed to refer to primary sources, we generally opted for the original research rather than quoting secondary sources. Following your feedback, we included more recent publications to our literature review to demonstrate the continuity of the discourse.

Kind regard, The Authors

Reviewer 3 Report

Thank you for providing me with the opportunity to review this paper. In general, I think the topic is interesting and relevant. At the same time, I do have some major issues with the content of the research questions and the set-up of the study. I believe that feeling pressured as a student may affect perceived autonomy support, this idea seems very interesting and relevant to me. However, I find it confusing that the authors treat school pressure as a trigger for controlling teacher behaviour on one hand, while on the other hand they argue that is only affects students’ perceptions of that behaviour. This unclarity, to a large extent undermines the strength of the theoretical background.

               Another main issue I have with this paper is that relatedness, competence, and autonomy are linked with different outcome-measures, I am not convinced by the authors’ argumentation that is makes sense. I would rather say that all three needs are important for all outcome-measures. I also am not convinced that it makes sense to examine mediating effects of competence and relatedness on the relation between perceived autonomy and outcome measures. I believe it could be interesting, but then I would have liked arguments why it would be interesting.

Author Response

Thank you for your review of our work and your detailed feedback, which contributed to the revision of the manuscript.

With regard to R3: “. In general, I think the topic is interesting and relevant. At the same time, I do have some major issues with the content of the research questions and the set-up of the study. I believe that feeling pressured as a student may affect perceived autonomy support, this idea seems very interesting and relevant to me. However, I find it confusing that the authors treat school pressure as a trigger for controlling teacher behaviour on one hand, while on the other hand they argue that is only affects students’ perceptions of that behaviour. This unclarity, to a large extent undermines the strength of the theoretical background.“ we have the following response:

Thank you for this helpful comment. Regarding this comment on students’ behavior as trigger for teachers controlling teaching, we would like to reply that, in our research, we did not emphasize on this issue. We cited Pelletier and colleagues (2002) who reported this reciprocal relationship in different publications, as well as other authors within this framework did (e.g. Koka, 2013). As our design did not allow for testing reciprocal relations, we only focused on the associations of perceived pressure on students’ outcomes. School pressure is here not regarded as the trigger for controlling behaviour, but rather a result from teachers’ intentions to control students. Thus, students are assumed to perceive this behaviour also as controlling, which should subsequently be negatively related to perceived autonomy. We integrated this into the theoretical background in order to make that clearer.

With regards to the following comment: “Another main issue I have with this paper is that relatedness, competence, and autonomy are linked with different outcome-measures, I am not convinced by the authors’ argumentation that is makes sense. I would rather say that all three needs are important for all outcome-measures. I also am not convinced that it makes sense to examine mediating effects of competence and relatedness on the relation between perceived autonomy and outcome measures. I believe it could be interesting, but then I would have liked arguments why it would be interesting.” we have the following reply:

As you stated, all three basic psychosocial needs are associated with favorable outcomes, which is linked to concepts of universality and essentiality of the Self-Determination Theory (Vansteenkiste et al., 2020). On the other hand, facing the criterion “distinct”, each need „concerns a distinct set of experiences and its emergence is not contingent upon or derivative from the frustration of other needs” (Vansteenkiste et al., 2020, p. 4) and the strength of association with outcomes variables differs according to the characteristics of the outcome (Deci & Ryan, 2016; Olafsen et al., 2021).

For this, research in SDT focusses on the identification of different sets of strategies to enhance a specific need (e.g. Kaplan & Madjar, 2017) as well as examining mediating for each need (Hsu et al., 2019) in order to determine the strenght of the association of BPNs with predicting and outcome variables. The analysis of distinct BPNs, can be considered as a strength, as due to high collinearity, several studies need to create an aggregated factor and implement need satisfaction as one factor in their analysis model, resulting in methodological and practical limitations. The relevance of the analysis of mediating effects is presented in section 1.2 as follows. “Indeed, this could be made more precisely: According to SDT, all three basic psychosocial needs are associated with favourable outcomes [38]. On the other hand, each need concerns a distinct set of experiences and the strength of association with related variables differs according to the characteristics of the outcomes [34,46]. Correspondingly, research in SDT focusses on the identification of different sets of strategies to enhance a specific need (e.g. [47]) as well as examining mediating for each need [48]. We integrated this within the introduction. “

Thank you again for your helpful comments to improve this manuscript!

Kind regard, The Authors

Deci, E. L., & Ryan, R. M. (2016). Optimizing students' motivation in the era of testing and pressure: A self-determination theory perspective. In W. C. Liu, J. C. K. Wang, & R. M. Ryan (Eds.), Building autonomous learners: Perspectives from research and practice using self-determination theory (pp. 9–29). Springer.

Hsu, H.‑C. K., Wang, C. V., & Levesque-Bristol, C. (2019). Reexamining the impact of self-determination theory on learning outcomes in the online learning environment. Education and Information Technologies, 24(3), 2159–2174. https://doi.org/10.1007/s10639-019-09863-w

Kaplan, H. & Madjar, N. (2017). The motivational outcomes of psychological need support among pre-service teachers: Multicultural and Self-determination Theory perspectives. Frontiers in Education, 2. https://doi.org/10.3389/feduc.2017.00042

Olafsen, A. H., Halvari, H. & Frølund, C. W. (2021). The Basic Psychological Need Satisfaction and Need Frustration at Work Scale: A Validation Study. Frontiers in Psychology, 12, 697306. https://doi.org/10.3389/fpsyg.2021.697306

Koka, A. (2013). The Relationships Between Perceived Teaching Behaviors and Motivation in Physical Education: A One-Year Longitudinal Study. In Scandinavian Journal of Educational https://doi.org/10.1080/00313831.2011.621213

Pelletier, L. G., Séguin-Lévesque, C., & Legault, L. (2002). Pressure from above and pressure from below as determinants of teachers' motivation and teaching behaviors. 

Round 2

Reviewer 2 Report

The results could be strengthened by evaluating the trends observed and connecting the significance of the results to wider understanding by referencing literature that support the results. 

Author Response

Thank you for evaluating the first revision and for providing additional feedback. We strengthen the discussion and added additional current literature to combine our findings with the scientific discourse.

Reviewer 3 Report

Regarding my first main concern (the position of feeling pressured), in the paper it now is clear that perceived pressure is treated as an element of controlling teaching behavior. I am wondering, however, how this adds to previous research. While the authors argue that their approach is innovative, I thought school pressure has always been considered an element of controlling teaching. It would be helpful if the authors could explain what makes their approach innovative. I felt the link the authors make with support for the other two needs (for relatedness and competence) could be the innovative part, not the fact that they examine school pressure I itself. In their summary (above limitations in the Discussion) it is about that: school pressure affecting all kinds of things. I feel that is the main point of the paper and an important conclusion. In the Introduction the authors argue that a lot of research has been done on what triggers school pressure. I wouldn’t agree with that, although some studies have been done. I think the type of SDT-research that has been most popular by far is research that links need support with motivation.

               Regarding my second main concern (linking each of the three needs with different outcome measures), it is clearer now why the authors expect strengths of associations to vary. However, I still miss argumentation on why it would be interesting to investigate this relationship.

Overall, I think the paper has improved and its focus is much clearer now.

Author Response

(The authors gave the same response as above.)

Round 3

Reviewer 3 Report

Unfortunately, I still cannot recommend publication of the paper in its current form. My main concern is that it still does not become very clear from the Introduction/Theoretical background why the paper is a relevant addition to current SDT-research. A very large number of SDT-studies is available already that links (aspects of) need-supportive teaching and student motivation, as the authors mention in their paper. The large majority of these studies have relied on student perceptions to measure need supportive teaching. Although I do believe this paper to be potentially relevant for the field, I am afraid I cannot recommend publication as long as its not clear why this paper is a relevant addition to the field. To make that clear major changes to the Introduction, as I tried to argue in my review of the prior version.

Most importantly, it still remains unclear how “school pressure” relates to prior research on autonomy support and thwart (one of the main concerns I expressed in my review of the prior version). The authors state that there is a lack of studies, but it is unclear where they base this statement on. Particularly, I miss information on how their study is different from other studies that were conducted. I believe this information to be key to understanding to what extent the present study is a relevant addition to prior SDT-research. Further, I still miss argumentation in the Introduction on why linking each of the three needs with different outcome measures is relevant. I do like that the authors have added to the Discussion that “ basic psychological needs were analysed separately allowing a deeper understanding of relations to relevant outcome variables”, but I think more specific information in the Introduction is needed as well. Specifically, how is the present study a relevant addition to prior research in this respect?

Author Response

Thank you for clarifying that one element of your review #2 was not fully addressed in our second revision. In our third revision, at the end of the study design, we added a statement concerning the added value of our research.

We like to thank you for your patience and support in the improvement of our manuscript.